# Mephedrone and Its Metabolites: A Narrative Review

**DOI:** 10.3390/ijms26157656

**Published:** 2025-08-07

**Authors:** Ordak Michal, Tkacz Daria, Juzwiuk Izabela, Wiktoria Gorecka, Nasierowski Tadeusz, Muszynska Elzbieta, Bujalska-Zadrozny Magdanena

**Affiliations:** 1Department of Pharmacotherapy and Pharmaceutical Care, Faculty of Pharmacy, Medical University of Warsaw, Banacha 1 Str., 02-097 Warsaw, Poland; s080971@student.wum.edu.pl (T.D.); s080879@student.wum.edu.pl (J.I.); s083172@student.wum.edu.pl (W.G.); magdalena.bujalska@wum.edu.pl (B.-Z.M.); 2Department of Psychiatry, Faculty of Pharmacy, Medical University of Warsaw, Nowowiejska 27 Str., 00-665 Warsaw, Poland; tadeusz.nasierowski@wum.edu.pl; 3Department of Medical Biology, Medical University of Bialystok, Mickiewicza 2c Str., 15-222 Bialystok, Poland; elzbieta.muszynska@umb.edu.pl

**Keywords:** mephedrone, new psychoactive substances, metabolites

## Abstract

New psychoactive substances (NPSs) have emerged as a significant global public health challenge due to their ability to mimic traditional drugs. Among these, mephedrone has gained attention because of its widespread use and associated toxicities. This review provides a comprehensive analysis of the structure, pharmacokinetic properties, and metabolic pathways of mephedrone, highlighting its phase I and phase II metabolites as potential biomarkers for detection and forensic applications. A comprehensive literature search was performed without date restrictions. The search employed key terms such as “mephedrone metabolites”, “pharmacokinetics of mephedrone”, “phase I metabolites of mephedrone”, and “phase II metabolites of mephedrone”. Additionally, the reference lists of selected studies were screened to ensure a thorough review of the literature. Mephedrone is a chiral compound existing in two enantiomeric forms, exhibiting different affinities for monoamine transporters and distinct pharmacological profiles. In vivo animal studies indicate rapid absorption, significant tissue distribution, and the formation of multiple phase I metabolites (e.g., normephedrone, dihydromephedrone, 4-carboxymephedrone) that influence its neurochemical effects. Phase II metabolism involves conjugation reactions leading to metabolites such as N-succinyl-normephedrone and N-glutaryl-normephedrone, further complicating its metabolic profile. These findings underscore the importance of elucidating mephedrone’s metabolic pathways to improve detection methods, enhance our understanding of its toxicological risks, and inform future therapeutic strategies.

## 1. Introduction

New psychoactive substances (NPSs) have emerged as a significant global public health issue. The prevalence of NPSs has grown globally, with their use reported in over 134 countries and territories. By December 2022, the UNODC Early Warning Advisory had listed more than 1124 NPS compounds, highlighting the rapid expansion and complexity of this challenge [1]. The United Nations Office on Drugs and Crime (UNODC) defines NPSs as substances not regulated under the 1961 Single Convention on Narcotic Drugs or the 1971 Convention on Psychotropic Substances, but which may pose health risks in their pure form or as part of a mixture. Although some of these substances were originally synthesized decades ago, their classification as “new” reflects their recent introduction to commercial markets, rather than their invention. These substances can cause a variety of health and social harms, and their toxicity is increasingly recognized in both clinical and public health settings [2].

### 1.1. Clinical Background and Relevance of Mephedrone

According to a report published in *The Lancet*, despite being classified as a controlled substance in the UK in 2010, mephedrone use has continued to rise, particularly in England, where an increasing number of individuals are seeking treatment for its abuse. Toxicology reports from London and Southeast England, cited in the publication, indicate the growing presence of mephedrone in post-mortem analyses, with related deaths increasing from 22 cases in 2014 to 34 cases in 2015. These fatalities span diverse user groups, including recreational drug users, intravenous drug users, and men who have sex with men (MSM), with MSM accounting for 68% of such deaths in 2015 [3]. Although regulations have aimed to curb its availability and usage, these data suggest that mephedrone remains a prevalent drug of choice within certain populations. Similar data were published in *The Lancet Psychiatry*, highlighting the severity of mephedrone abuse in Poland. Between August 2010 and April 2018, 601 individuals were admitted to Nowowiejski Hospital in Warsaw following prolonged mephedrone use (two or more consecutive days), with men comprising 93% of these cases. The average binge duration was 12.4 days, and the average daily dose reached 1.52 g, with younger individuals consuming higher doses and engaging in longer binges. The number of hospital admissions related to mephedrone binges steadily increased during this period, emphasizing the growing impact of this drug in Central Europe [4]. Previous publications have primarily focused on the psychiatric characterization of case reports and larger groups of patients who misuse this class of novel psychoactive substances (NPSs) [5].

### 1.2. Importance of Metabolite Research

Additionally, studies on the pharmacokinetics of mephedrone and its metabolites have been continuously published; however, a comprehensive review in this area remains lacking. The analysis of NPS metabolites provides crucial insights into biotransformation processes, including oxidation and reduction reactions (phase I), as well as conjugation with glucuronic or sulfuric acid (phase II). This knowledge enhances our understanding of the substance’s mechanism of action within the body and its potential toxic effects. Moreover, metabolites can serve as diagnostic biomarkers, enabling the detection of prior NPS use even after the parent compound is no longer present in the body. Studies also suggest that some metabolites may exhibit higher toxicity than the parent compound itself, emphasizing the need for further research in this area. Understanding metabolic pathways and metabolite profiles could also contribute to the development of more effective treatment strategies and more sensitive diagnostic tests, both of which are essential for forensic and clinical toxicology [6,7,8,9]. This review adopts a metabolism-centered perspective, aiming to provide a comprehensive synthesis of mephedrone’s phase I and phase II biotransformation pathways and their clinical, toxicological, and forensic implications. By focusing on metabolic processes as the organizing framework, we aim to bridge gaps between pharmacokinetics, stereochemistry, analytical toxicology, and public health monitoring. To our knowledge, no previous review has taken this integrative, metabolism-driven approach to understanding mephedrone’s impact.

### 1.3. Aim of Review and Methodology

Therefore, the aim of this review was to analyze the existing literature on the structure, pharmacokinetic properties, and phase I and phase II metabolites of mephedrone. This review also serves as a step toward identifying areas requiring further investigation, which may be critical in developing regulatory and preventive strategies related to mephedrone use. A comprehensive search was conducted using international databases including Web of Science, PubMed/MEDLINE, ScienceDirect, Scopus, and Google Scholar to identify articles related to mephedrone metabolites. The search terms used were “mephedrone metabolites”, “pharmacokinetics of mephedrone”, “phase I metabolites of mephedrone”, and “phase II metabolites of mephedrone”. All publication dates and languages were considered, and article selection was based on relevance to the topic. As this is a narrative review, no formal exclusion criteria were applied, and studies were not filtered by study design or evaluated using standardized quality assessment tools. Instead, articles were selected based on their relevance to the scope of the review. Additional references were identified by manually screening the bibliographies of selected studies. In total, 31 references were included. Of these, the first nine were used to illustrate the clinical importance and growing problem of mephedrone abuse, particularly the rising number of hospitalizations, while the remaining sources focused specifically on the pharmacokinetics and metabolism of mephedrone, including phase I and phase II pathways.

## 2. Structure and Pharmacokinetic Properties of Mephedrone

Mephedrone (4-methylmethcathinone, 4-MMC) is structurally characterized by the presence of a chiral center at the α-carbon. It exists in two enantiomeric forms. Both enantiomers exhibit similar affinity for the dopamine transporter (DAT); however, (S)-mephedrone is 50 times more potent as a mediator of serotonergic neurotransmission, whereas (R)-mephedrone is primarily responsible for its euphoric effects [10]. In in vivo pharmacokinetic studies of mephedrone conducted on rats, the mean maximum plasma concentration of mephedrone was reached within 30 min. The compound’s penetration into the brain was not significantly delayed compared to plasma levels, as the mean peak concentration in brain tissue was also observed 30 min after administration. Mephedrone exhibited significant accumulation in the lungs, where its concentration at 30 min exceeded those in the plasma, brain, and liver. Four hours post-administration, the presence of the compound in plasma and all tissues was nearly undetectable [11].

Mephedrone demonstrates lower hydrophobicity compared to other amphetamine-like compounds, such as 3,4-methylenedioxy-methamphetamine (MDMA, ecstasy) and methamphetamine, resulting in a reduced ability to cross the blood–brain barrier. In a study on rats administered mephedrone orally (doses: 30 mg, 60 mg) or intravenously (dose: 10 mg/kg body weight), the plasma protein binding percentage was reported to be 21.59 ± 3.67% [12].

Mephedrone undergoes hepatic metabolism. In an in vitro study, the exposure of mephedrone to cytochrome P450 isoforms 1A2, 2B6, 2C8, 2C18, 2C19, and 3A4 led to the formation of its metabolites, which were identified using ultra-performance liquid chromatography coupled with time-of-flight mass spectrometry (UPLC-TOF/MS) in positive ionization mode. However, the activity of these isoforms did not result in a significant reduction in mephedrone levels. Furthermore, assessments of their activity in the presence of inhibitors (e.g., furafylline, omeprazole, ketoconazole) indicated no substantial changes in substrate depletion, suggesting that CYP1A2, 2B6, 2C8, 2C18, 2C19, and 3A4 do not play a significant role in mephedrone metabolism at the tested concentrations [13].

In contrast, the use of quinidine, a CYP2D6 inhibitor, demonstrated a pronounced effect on substrate depletion, with only 20% consumption over 140 min, compared to 60% in the absence of the inhibitor within the same timeframe. This finding confirms that mephedrone is primarily metabolized via CYP2D6 activity [13]. Mephedrone undergoes metabolism through three primary pathways involving CYP2D6: (1) N-demethylation, (2) hydroxylation, and (3) reduction of the β keto group. The N-demethylation of mephedrone leads to the formation of normephedrone, while reduction results in the formation of dihydromephedrone. Through the oxidation of its alkyl group to a carboxyl group, mephedrone first transforms into 4-hydroxymephedrone. Subsequently, the hydroxyl group undergoes further oxidation, yielding 4-carboxymephedrone. Additionally, normephedrone, dihydromephedrone, and their derivatives can also undergo oxidation of the alkyl group to a carboxyl group, further contributing to mephedrone’s metabolic profile [14].

## 3. Preclinical Studies

### 3.1. Metabolism and Pharmacokinetics

In in vivo rat urine samples, analyzed using liquid chromatography coupled with a Q-Exactive system (LC-QE), the following metabolites were identified: 4-hydroxymephedrone, 4-carboxymephedrone, 4-carboxynormephedrone, 3O-glucuronide of hydroxymephedrone, and glucuronide of normephedrone. The comparison of in vitro mephedrone metabolism in primary human liver microsomes (pHLMs) with the results from a human pooled urine analysis and in vivo rat urine analysis revealed a notable stereoselective difference. While S-(−)-mephedrone was the predominant excreted form in both pHLM and human urine samples, R-(+)-mephedrone was the primary form detected in rat urine. The researchers hypothesized that mephedrone metabolism in humans and rats may follow stereoselectively distinct pathways, raising concerns about the reliability of the rat model as the sole approach to analyzing mephedrone metabolites or potential biomarkers [15].

### 3.2. Neurochemical and Behavioral Effects

Animal studies have demonstrated that mephedrone induces the release of monoamines—specifically noradrenaline and dopamine—by inhibiting monoamine transporters, primarily the noradrenaline transporter (NAT) and dopamine transporter (DAT). Its effect on the serotonin transporter (SERT) is comparatively weaker. Mephedrone also promotes the release of 5-hydroxytryptamine (5-HT, serotonin), which acts on 5-HT1B and 5-HT2 receptors, both of which contribute to its stimulating effects [16,17,18]. Although the impact of mephedrone on monoamine transporters has been well studied and characterized, knowledge regarding its effects on glutamatergic neurotransmission remains limited. Glutamatergic signaling plays a crucial role in the reward system and the addiction mechanisms of various substances of abuse, including heroin, cocaine, nicotine, and alcohol. Behavioral studies using a conditioned place preference (CPP) test—a widely used Pavlovian conditioning-based method for the assessment of drug rewards in a rat model—have demonstrated that mephedrone exhibits rewarding properties, at least partially mediated by alterations in glutamatergic neurotransmission. Furthermore, findings indicate that mephedrone-induced changes in glutamate levels are long-lasting, persisting for up to two weeks after six consecutive days of drug administration [19].

In rats administered mephedrone orally (doses: 30 mg, 60 mg) or intravenously (dose: 10 mg/kg body weight), a significant increase in locomotor activity was observed. Following oral administration, the level of stimulation was found to be dose-dependent. After intravenous administration, this effect persisted for 120 min, whereas, after the oral administration of 60 mg, the effect lasted for 90 min. In the same study, the high steady-state volume of distribution (Vss) indicated that mephedrone undergoes extensive tissue distribution in rats [12].

Studies using a rat model indicate that repeated mephedrone exposure during adolescence leads to impairments in spatial memory and cognitive function in adulthood [19]. This effect may be associated with the increased expression of matrix metallopeptidase 9 (MMP-9), which was observed in the hippocampus and piriform cortex in rats following mephedrone withdrawal. Under normal conditions, MMP-9 is activated at the synapse through β1-integrin and plays a regulatory role in N-methyl-D-aspartate (NMDA) receptor function. However, excessive MMP-9 expression enhances basal excitatory neurotransmission while reducing synaptic plasticity, which may contribute to the observed cognitive deficits [20].

### 3.3. Thermoregulatory Response

In a rat model, a statistically significant increase in body temperature was observed in animals administered mephedrone at doses of 20 mg/kg, with temperature elevations of up to 1 °C. Similarly, in a mouse model, mephedrone induced transient hyperthermia. Notably, the duration of elevated body temperature (up to 3 h post-administration) did not align with mephedrone’s rapid pharmacokinetic profile, suggesting that additional factors, such as active metabolites, may contribute to the prolonged effect [11]. The authors hypothesized that the observed hyperthermia could have been linked to the overall increase in locomotor activity due to dopaminergic stimulation. However, since locomotor excitation did not persist as long as the hyperthermic effect, the exact mechanisms remain unclear. This discrepancy highlights the need for further research to elucidate the underlying processes [16].

## 4. Detection of Mephedrone in Biological Samples

Mephedrone, like other new psychoactive substances (NPSs), can be detected and quantified in biological samples, including whole blood, plasma, serum, and urine. Analytical techniques commonly used for NPS detection in biological matrices include liquid chromatography–mass spectrometry (LC-MS), liquid chromatography–tandem mass spectrometry (LC-MS/MS), gas chromatography–mass spectrometry (GC-MS), and gas chromatography–tandem mass spectrometry (GC-MS/MS) [10]. For example, GC-MS allows for the simultaneous detection of multiple psychoactive substances from different chemical groups in whole blood without requiring derivatization [21]. However, it is important to note that cathinone derivatives are thermally degradable, meaning that, under the high-temperature conditions required for GC-based analysis, these compounds may undergo partial decomposition, potentially affecting the accuracy of their quantification. For the isolation of NPSs from biological samples, liquid–liquid extraction (LLE) and solid-phase extraction (SPE) are commonly used techniques. LLE is a simple method; however, its drawbacks include a high risk of sample contamination and matrix effects that may influence the results. In contrast, SPE offers greater selectivity but is more time-consuming [10]. Other extraction methods include solid-phase microextraction (SPME), dispersive liquid–liquid microextraction (DLLME), and molecularly imprinted solid-phase extraction (mi-SPE). For the rapid detection of psychoactive substances, immunological methods can be utilized. However, no immunoassay is currently available to differentiate cathinone analogs, limiting their application in distinguishing mephedrone from its structural derivatives [10]. The detection of mephedrone in biological samples may be enhanced by utilizing its metabolites as biomarkers. Studies analyzing the presence of mephedrone and its metabolites in wastewater have identified several compounds, including 1-dihydromephedrone, N-sulfate normephedrone, 4-hydroxynormephedrone, 4-carboxymephedrone, 4-carboxynormephedrone, and 1-dihydronormephedrone. Mephedrone itself serves as a valuable biomarker due to its stability in wastewater. Even after 24 h at 4 °C and one week of incubation in wastewater, no significant metabolite formation was observed, suggesting that mephedrone remains chemically stable in such environmental conditions [15].

## 5. Phase I Metabolites

### 5.1. Normephedrone (4-MC)

Normephedrone (4-methylcathinone, 4-MC) is the primary metabolite of mephedrone, formed through N-demethylation [22] (Figure 1, reaction 3). This metabolite exhibits activity at the dopamine transporter (DAT) and noradrenaline transporter (NAT) but has significantly weaker interactions with the serotonin transporter (SERT) [14]. Normephedrone exists in two stereoisomeric forms: S-normephedrone (S-4-MC) and R-normephedrone (R-4-MC). Both enantiomers act as DAT and NAT inhibitors, exhibiting similar inhibitory activity. However, the S-enantiomer is up to 18 times more potent than the R-enantiomer in promoting serotonin release in SERT-expressing cells. Following hepatic transformation, 4-MC crosses the blood–brain barrier, facilitated by its partition coefficient (logD = 1.29), which, along with its molecular size, enhances the probability of brain transport [23]. The half-life of 4-MC is approximately 4.5 h, and it is primarily excreted in the urine. In human urine, S-4-MC is present at nearly twice the concentration of R-4-MC, indicating that 4-MC is predominantly excreted as the S-enantiomer [24]. While 4-MC follows a similar elimination profile to mephedrone, its concentration declines more gradually—in contrast to the sharp drop observed for mephedrone immediately after reaching peak levels. This results in a slight delay in 4-MC elimination and higher concentrations of 4-MC in both the brain and serum compared to mephedrone at three hours post-administration. This difference may be attributed to the slightly higher polarity of 4-MC, which reduces its rate of blood–brain barrier penetration. Theoretically, this suggests that 4-MC may contribute to the prolonged or delayed effects associated with mephedrone administration [11]. Similar findings were obtained in a study on mephedrone metabolism and metabolite distribution in a rat model. One hour after the intraperitoneal administration of mephedrone at a dose of 30 mg/kg body weight, the 4-MC concentrations exceeded the mephedrone levels by 100%, with Cmax values of 179 ± 29.3 ng/mL for 4-MMC and 54.4 ± 4.9 ng/mL for 4-MC. The authors highlighted that, given the dose used in the study (150 mg), it can be postulated that brain concentrations would fall within the low micromolar range following moderate and high doses of mephedrone. These findings support the notion that mephedrone’s effects may be significantly influenced by its metabolites, particularly normephedrone [24]. Normephedrone undergoes further biotransformation into phase II metabolites through conjugation with succinyl-coenzyme A, glutaryl-coenzyme A, and adipoyl-coenzyme A. According to Linhart et al., normephedrone is the first known xenobiotic to undergo conjugation with dicarboxylic acids in mammals [22]. In vitro studies using animal models have compared the effects of mephedrone and its metabolites on monoamine uptake and release via transporters in human DAT-, NAT-, and SERT-expressing cells, as well as in rat brain synaptosomes. These studies demonstrated that phase I metabolites act as substrates for monoamine transporters. However, only 4-MC appears to exert significant neurochemical effects in vivo, potentially contributing to the behavioral effects observed following systemic mephedrone administration. In vitro findings showed that mephedrone increased the extracellular dopamine levels following systemic doses of 1 mg/kg and 3 mg/kg, while 4-MC produced a similar effect at 3 mg/kg. The neurochemical effects of systemically administered mephedrone and 4-MC at various doses were further investigated in vivo using microdialysis in the rat caudate nucleus. Both mephedrone and 4-MC elevated the serotonin levels in the dialysate, with 15-fold and 25-fold increases above the baseline values, respectively, for 1 mg/kg and 3 mg/kg doses. Additionally, mephedrone and 4-MC significantly influenced movement-related behaviors, as both compounds stimulated locomotor activity in rats [23]. Behavioral studies in rats revealed that the stimulatory effect of 4-MC on locomotor activity diminished rapidly, consistent with the observed tissue and serum concentrations in the study. The degree of stimulation in rats administered 4-MC was comparable to that observed in rats given mephedrone. The highly transient stimulant effect of mephedrone may contribute to frequent redosing in humans, potentially increasing the risk of addiction [11].

An in vitro receptor-binding profile study of various mephedrone metabolites utilized radioligand-binding assays to assess the affinity for trace amine-associated receptor 1 (TAAR1). S-4-MC exhibited submicromolar affinity for rat and mouse TAAR1 receptors. A separate in vitro study in a human model, also employing radioligand-binding assays, evaluated the affinity of different metabolites for receptors such as α1A, 5-HT2A, 5-HT2B, 5-HT2C, and TAAR1. The results indicated that none of the examined metabolites bound to 5-HT1A or 5-HT2C, nor did they activate 5-HT2B receptors. However, S-4-MC bound to the human α1A and 5-HT2A receptors, exhibiting supramicromolar affinity. Its agonistic activity at α1A receptors suggests a stimulatory effect, including enhanced locomotor activity. Additionally, its direct sympathomimetic action appears to synergize with its well-established indirect sympathomimetic effects. The interaction with 5-HT2A receptors may partially contribute to the hallucinogenic effects associated with mephedrone use. Moreover, S-4-MC demonstrated submicromolar affinity for human TAAR1 receptors, which play a self-inhibitory role in monoaminergic neurons. This suggests a potential neuroprotective effect, although the precise role of TAAR1 activation in psychostimulant effects remains unclear [24]. Studies on mephedrone and its metabolites in wastewater, compared with in vitro findings, revealed that 4-MC cannot be used as a biomarker of mephedrone consumption, despite its stereoselective formation in pHLM in vitro. 4-MC exhibited low stability and was not detectable in wastewater samples or in pooled human urine samples, suggesting that it undergoes rapid degradation or further metabolic transformation before excretion [15].

### 5.2. Dihydromephedrone (DHMMC)

Another important phase I metabolite is dihydromephedrone (DHMMC), which is formed through the reduction of the carbonyl group to a hydroxyl group (Figure 1, reaction 1). This metabolite exhibits activity at the DAT and NAT but has a significantly weaker effect on the SERT [14]. Dihydromephedrone exists in four stereoisomeric forms: (1R,2R)-dihydromephedrone, (1S,2S)-dihydromephedrone, (1S,2R)-dihydromephedrone, and (1R,2S)-dihydromephedrone.

The half-life of DHMMC is 5.7 h [24]. An in vitro study comparing the effects of mephedrone and its metabolites on monoamine uptake and release via transporters in human DAT-, NAT-, and SERT-expressing cells and rat brain synaptosomes indicated that phase I metabolites act as substrates for monoamine transporters. However, 4-MC and 4-OH-MMC exhibited significantly stronger activity in this regard than DHMMC. In vivo administration in rats demonstrated that, unlike mephedrone and 4-MC, DHMMC did not increase the extracellular dopamine levels or stimulate locomotor activity in the tested animals [23].

An in vitro receptor-binding profile study of various mephedrone metabolites, using radioligand-binding assays, examined their affinity for TAAR1 receptors. The 1R,2R-DHMMC, 1S,2S-DHMMC, and 1R,2S-DHMMC isomers bound to TAAR1 receptors in rat cells, but only 1S,2S-DHMMC bound to these receptors in mouse cells. Within the tested concentration range (EC50 > 30 μM), no activity was observed at TAAR1 receptors in human cells. Additionally, 1S,2S-DHMMC and 1R,2S-DHMMC interacted with 5-HT2A receptors, while only 1R,2S-DHMMC exhibited affinity for α2A receptors [24].

Current in vitro data from studies conducted on human HEK293 kidney cells indicate that mephedrone metabolites inhibit monoamine uptake in a concentration-dependent manner across all three membrane monoamine transporters (hDAT, hNAT, hSERT). 4-MC and 4-OH-MMC inhibit uptake at hDAT and hNAT with potency comparable to mephedrone, whereas DHMMC exhibited significantly weaker effects [23]. This metabolite is detectable in biological samples, making it a potential biomarker of mephedrone consumption, based on wastewater analysis studies [15].

### 5.3. Dihydronormephedrone (DHMC)

Another phase I metabolite is dihydronormephedrone (DHMC). DHMC is formed from 4-MMC through the N-demethylation of the secondary amine, followed by the reduction of the carbonyl group (Figure 1, reaction 4) [25]. DHMC is characterized by a renal clearance rate of 252 ± 294 mL/min, as observed in a study conducted six hours after mephedrone administration in six individuals with prior experience using mephedrone or other stimulants. Following intranasal administration, the metabolite remained detectable in urine samples for up to three days [26,27]. Compared to other metabolites, such as 4-COOH-MC or the parent compound, which were detected in urine at Cmax concentrations of 29.8 μg/mL and 6.98 μg/mL, respectively, DHMC reached significantly lower concentrations (Cmax = 93.1 ng/mL). This metabolite may serve as a biomarker of mephedrone consumption, particularly due to its prolonged detectability in urine samples [15,26].

### 5.4. 4-Carboxymephedrone (4-COOH-MMC)

Another relatively well-studied phase I metabolite is 4-carboxymephedrone (4-COOH-MMC), which is formed from mephedrone in two steps—hydroxylation followed by the oxidation of the methyl group on the aromatic ring [22] (Figure 1, reactions 15 and 16). A metabolomic study using LC-HRMS indicated that, three hours after intranasal administration, both the mephedrone and 4-carboxymephedrone levels in plasma remained high, while, after six hours, both compounds were detectable only in trace amounts [28]. Czerwińska et al. demonstrated that, after the intranasal administration of 100 mg mephedrone, 4-COOH-MMC could be detected in whole blood and plasma within 5–15 min. Interindividual differences in detection were likely due to CYP2D6 polymorphism, as CYP2D6 is the primary CYP450 isoenzyme responsible for mephedrone metabolism. In terms of elimination kinetics, mephedrone and 4-COOH-MMC exhibited similar pharmacokinetic parameters [29]. 4-COOH-MMC was one of the most abundant metabolites detected in human plasma, with an AUC_0_–_8_h value of 113% following the oral consumption of 150 mg of mephedrone (n = 6 individuals) [24]. Two studies analyzing urinary excretion following the oral administration of 150 mg of mephedrone, using LC-MS/MS and GC-MS, demonstrated that 4-COOH-MMC was excreted in urine at concentrations up to 10 times higher than mephedrone [30,31]. Similarly, after the intranasal administration of 100 mg of mephedrone hydrochloride, the 4-carboxymephedrone concentrations in urine significantly exceeded those of the parent compound. Additionally, the recovered urinary concentrations of 4-COOH-MMC were approximately 10 times higher than those of mephedrone, and the metabolite remained detectable in urine for up to three days post-administration [27]. The racemic mixture of 4-COOH-MMC showed no activity at the DAT, NAT, or SERT in an in vitro study conducted on HEK293 cell lines [14]. 4-COOH-MMC was detected in rat blood using LC/MS following the oral administration of mephedrone (30 mg, 60 mg) or intravenous administration (10 mg/kg body weight), with measurements taken 30, 60, and 120 min post-administration. One hour after the intraperitoneal administration of mephedrone (30 mg/kg), 4-COOH-MMC was detectable in small amounts in the rat prefrontal cortex [30]. In vivo forensic urine and blood samples have confirmed the presence of 4-carboxymephedrone [13]. Additionally, this metabolite was detected in wastewater samples, and researchers suggest that it could serve as a biomarker of mephedrone abuse [15].

### 5.5. 4-Hydroxymephedrone (4-OH-MMC)

Another phase I metabolite is 4-hydroxymephedrone (4-OH-MMC), which is formed through the hydroxylation of 4-MMC at the 4-position [30] (Figure 1, reaction 15). 4-OH-MMC exists in two stereoisomeric forms: S-4-OH-mephedrone (S-4-OH-MMC) and R-4-OH-mephedrone (R-4-OH-MMC). Both enantiomers function as inhibitors of the DAT, NAT, and SERT, but the S-enantiomer exhibits greater activity. The most significant difference in enantiomeric activity is observed in SERT inhibition, where the S-enantiomer is 19 times more potent than the R-enantiomer [24]. Data from in vitro studies conducted on human HEK293 kidney cells indicate that mephedrone metabolites inhibit monoamine uptake in a concentration-dependent manner at the hDAT, hNAT, and hSERT. 4-OH-MMC inhibits uptake at the hDAT and hNAT with potency comparable to mephedrone [23]. A receptor-binding profile study analyzing mephedrone and its derivatives demonstrated that 4-OH-MMC does not bind to 5-HT1A or 5-HT2C receptors and does not activate the 5-HT2B receptor.

The next phase I metabolite is hydroxytolyldihydronormephedrone (4-OH-DH-MC), which is formed through three sequential reactions: mephedrone first undergoes oxidative N-demethylation to form normephedrone, which is then hydroxylated at the 4-methyl group to produce 4-hydroxynormephedrone. Finally, the carbonyl group of 4-hydroxynormephedrone is reduced (Figure 1, reactions 3, 13, and 14) [22].

### 5.6. Other Phase I Metabolites

4-OH-DH-MC was first described by Linhart et al., who classified it as a minor metabolite due to its low percentage in rat urine (0.2 ± 0.1%). 4-OH-DH-MC was detected through detailed ion chromatogram analysis using HPLC/HRMS in rat urine samples following a single subcutaneous bolus dose of 20 mg/kg mephedrone [22]. To the best of our knowledge, no studies on this metabolite have been conducted in humans. Additionally, 4-carboxynormephedrone (4-COOH-MC) and 4-hydroxynormephedrone (4-OH-MC) remain poorly studied metabolites. 4-OH-MC is formed through the hydroxylation of the 4-methyl group of normephedrone, whereas 4-COOH-MC results from the carboxylation of the same group (Figure 1, reactions 9 and 13). The precise pharmacodynamic parameters and effects of these metabolites have not yet been studied. However, they are potential candidates for biomarkers of mephedrone consumption [15]. According to the literature, two additional phase I metabolites have been identified: normephedrone-ω-carboxylic acid (3-OOH-MC) and 4-carboxydihydromephedrone (4-COOH-DH-MMC). 3-OOH-MC is formed through the oxidation of the intermediate compound INT2, which is produced during the hydroxylation of 4-MC (Figure 1, reactions 10 and 11). 4-COOH-DH-MMC is formed following the reduction of the carbonyl group in 4-COOH-MMC (Figure 1, reaction 18) [22]. Beyond reports confirming their existence and describing the reactions leading to their formation, no further studies have been conducted on these two compounds.

## 6. Phase II Metabolites

### 6.1. N-Succinyl-Normephedrone (4-MC-SC)

One of the most extensively studied phase II metabolites is N-succinyl-normephedrone (4-MC-SC), which is formed through the oxidative N-demethylation of mephedrone, followed by the conjugation of the resulting normephedrone with succinyl-CoA (Figure 1, reaction 5) [22,30]. The half-life of N-succinyl-normephedrone (4-MC-SC) is 8.2 ± 2.6 h [24]. A pharmacokinetic study of 4-MMC and its metabolites, conducted using LC-MS/MS, revealed that 4-MC-SC formation was delayed compared to other metabolites. Its T_max_ was 3.7 h, whereas the Tmax for other metabolites ranged from 1.2 to 1.7 h. This delay is attributed to 4-MC-SC being formed from 4-MC rather than directly from 4-MMC. The study also evaluated the blood–brain barrier permeability for the analyzed compounds. 4-MC-SC reached measurable concentrations in brain tissue, with 7 ng/g detected one hour after drug administration. The metabolite was excreted approximately 8.2 h after mephedrone administration [30].

### 6.2. N-Glutaryl-Normephedrone (4-MC-GL)

Another phase II metabolite is N-glutaryl-normephedrone (4-MC-GL), which is formed from 4-MMC in two steps: first, through the oxidative N-demethylation of mephedrone, followed by the conjugation of normephedrone with glutaryl-CoA (Figure 1, reaction 7) [22,30]. This metabolite was detected in rat urine following mephedrone administration, with its presence confirmed via mass spectrometry analysis using total ion current (TIC) chromatograms, retention time comparisons, and HRMS and MS2 spectral analysis. 4-MC-GL is excreted in urine, representing a small fraction (2.3 ± 0.3%) of the total excreted compounds in rats administered a single 20 mg/kg subcutaneous bolus dose of mephedrone [22].

### 6.3. N-Adipoyl-Normephedrone (4-MC-AD)

Another phase II metabolite, N-adipoyl-normephedrone (4-MC-AD), is formed from 4-MC through two sequential reactions: first, the oxidative N-demethylation of mephedrone to normephedrone, followed by its transformation into 4-MC-AD (Figure 1, reactions 3 and 6). The likely precursor of this conjugation reaction is adipoyl-CoA, which is naturally present in the livers and kidneys of rats. Adipoyl-CoA, along with other acyl-coenzymes, acts as an active metabolic intermediate that is further conjugated with amino acids to form N-acylamino acids. N-adipoyl-normephedrone was first described by Linhart et al. after trace amounts (0.4 ± 0.2%) were detected in rat urine following a single subcutaneous bolus dose of 20 mg/kg mephedrone. The dicarboxylic acid conjugates 4-MC-SC, 4-MC-GL, and 4-MC-AD represent a novel class of phase II metabolites in mammals. Further research is needed to determine whether this phase II metabolic pathway also applies to other psychoactive amines and whether it plays a role in other mammalian species, including humans. Dicarboxylic acid amides derived from psychoactive amines may be highly useful as haptens in the development of immunochemical methods for drug abuse detection [22]. To date, none of these metabolites have been studied in pharmacokinetic or clinical trials, and their effects on the human body remain unknown.

### 6.4. Glucuronide Conjugates

Hydroxylmephedrone-3-O-glucuronide (OH-MMC-3-O-GLUC), hydroxylnormephedrone-3-O-glucuronide (OH-MC-3-O-GLUC), and 4-carboxymephedrone-N-glucuronide (4-COOH-MMC-N-GLUC) are phase II metabolites, with molecular formulas C_17_H_23_NO_8_, C_16_H_20_NO_8_, and C_17_H_21_NO_9_, respectively. These metabolites were detected and identified by Pozo et al. [25]. The proposed structures of these metabolites were determined based on ultra-high-performance liquid chromatography coupled with quadrupole time-of-flight mass spectrometry (UHPLC-QTOF MS). OH-MMC-3-O-GLUC is likely formed through the O-glucuronidation of an intermediate metabolite generated during mephedrone hydroxylation (Figure 1, reactions 19, 20). OH-MC-3-O-GLUC is produced via the O-glucuronidation of an intermediate metabolite formed during normephedrone hydroxylation (Figure 1, reactions 10 and 12). Meanwhile, 4-COOH-MMC-N-GLUC is generated through the N-glucuronidation of 4′-carboxymephedrone (Figure 1, reaction 17).

Normephedrone-N-glucuronide (4-MC-N-GLUC) is another phase II metabolite of mephedrone, identified in an in vitro study in which mephedrone was incubated in pooled human liver microsomes (pHLM) to generate both phase I and phase II metabolites. The study utilized untargeted LC-QTOF screening, employing the MetID software (https://www.acdlabs.com) to predict and detect a range of metabolites, including 4-MC-N-GLUC. The researchers did not discuss the potential use of this metabolite as a biomarker or the mechanism of its formation. However, based on its chemical structure, they proposed that N-glucuronidation is the reaction responsible for its formation (Figure 1, reaction 8) [15].

### 6.5. Normephedrone-N-Sulfate (4-MC-N-SUL)

The last identified phase II metabolite is normephedrone-N-sulfate (4-MC-N-SUL). A study on mephedrone and its metabolites in wastewater detected the presence of 4-MC-N-SUL. However, no properties of this metabolite were investigated, and the reaction leading to its formation was not provided. Furthermore, the researchers did not assess the potential of 4-MC-N-SUL as a biomarker of mephedrone use [15]. Based on its chemical structure, the authors suggested that N-demethylation followed by sulfonation leads to the formation of 4-MC-N-SUL (Figure 1, reaction 2). To the best of our knowledge, no further studies have been conducted on phase II metabolites in animals or humans regarding their presence, effects, pharmacokinetic properties, or potential applications. Figure 1 illustrates the metabolic pathway of mephedrone as inferred from the current literature.

## 7. Discussion

### 7.1. Metabolic Complexity and Pharmacological Implications

This review provides a comprehensive analysis of the structural, pharmacokinetic, and metabolic characteristics of mephedrone, highlighting the complexity of its biotransformation and its implications for clinical and forensic toxicology. Mephedrone, a chiral compound, exists as two enantiomers with distinct pharmacological properties. Both enantiomers inhibit monoamine transporters, but they differ markedly in their potency and neurochemical effects [14]. Preclinical studies in animal models have demonstrated rapid absorption, extensive tissue distribution, and swift metabolism resulting in several phase I metabolites, such as normephedrone, dihydromephedrone, and 4-carboxymephedrone. The stereoselective formation of these metabolites, with species-specific variations, underscores the need for caution when extrapolating these findings to human metabolism [11,15]. While individual studies describe stereoselectivity in isolation, few integrate these effects with broader clinical outcomes; this gap weakens the translational applicability. Moreover, despite frequent reference to CYP2D6-mediated metabolism, studies diverge regarding the extent to which genetic polymorphisms influence metabolite ratios—a discrepancy that limits predictive use in forensic settings. This differential activity of enantiomers, combined with species-specific metabolism, may partly explain the variability in clinical presentations and complicate toxicological interpretation. The absence of stereoselective detection protocols in routine screening may therefore lead to the under- or misestimation of mephedrone-related harm.

### 7.2. Integration of Metabolism with Analytical and Forensic Frameworks

This review further details the sequential metabolic processes, where phase I reactions including N-demethylation, reduction, and hydroxylation set the stage for subsequent phase II conjugation reactions [21]. The formation of conjugated metabolites, including N-succinyl-normephedrone, N-glutaryl-normephedrone, and N-adipoyl-normephedrone, provides a more stable metabolic profile, and they may serve as reliable biomarkers for mephedrone exposure in both biological and environmental matrices [22]. However, no consensus exists regarding which of these conjugates are most robust across populations or sample types. The current data are derived largely from in vitro or rodent studies, and their cross-study reproducibility is limited by analytical variability and inconsistent metabolite quantification. Advanced analytical techniques, such as LC-MS/MS and GC-MS/MS, have proven effective in detecting mephedrone and its metabolites [30], although challenges remain regarding the thermal degradation of cathinone derivatives during GC-based analyses [31]. The continued reliance on GC-MS, despite its susceptibility to degradation artifacts, reflects a methodological inertia that undermines the accuracy of forensic assessments. This highlights a tension between methodological tradition and the need for innovation in metabolite-specific diagnostics.

### 7.3. Environmental and Public Health Relevance of Metabolic Pathways

Recent studies have further enriched our understanding of mephedrone’s toxicodynamics and environmental impacts. In vivo studies in animal models have revealed that specific phase I metabolites may exhibit increased affinity for monoamine transporters, which could account for the prolonged stimulant effects observed following mephedrone administration [11,14]. Moreover, behavioral studies have indicated that repeated mephedrone exposure leads to lasting alterations in neurochemical profiles and neuroinflammatory responses, potentially contributing to cognitive deficits observed later in life [19]. However, neurotoxicity data remain inconsistent. Some studies report dopaminergic resilience post-exposure, while others observe functional deficits, suggesting a need for unified endpoints and longitudinal designs. Complementary investigations have also demonstrated that mephedrone and its metabolites can be reliably detected in wastewater, supporting their utility as biomarkers for the monitoring of community-level drug use [15]. The integration of metabolite-specific wastewater data with clinical toxicology could provide early-warning indicators of shifts in drug use patterns or metabolism-modulating factors (e.g., co-use with CYP inhibitors). This type of translational surveillance remains underutilized. The data presented in this review underscore the necessity of bridging multiple disciplines to fully understand mephedrone’s public health impacts. For example, stereoselective pharmacological effects influence metabolic pathways shaped by CYP2D6 polymorphism, which in turn dictates both clinical presentation and detectability in forensic analyses. Analytical limitations, such as thermal degradation in GC-MS, further complicate the interpretation of pharmacokinetic findings. Only by integrating pharmacology, toxicokinetics, stereochemistry, and analytical chemistry can a coherent and clinically relevant picture of mephedrone metabolism emerge. Altogether, this metabolism-centered synthesis not only maps the biotransformation of mephedrone but also integrates its implications across pharmacology, toxicology, diagnostics, and public health monitoring. This integrative approach provides not only a descriptive catalog of mephedrone metabolites but also a critical framework for understanding how metabolic transformations shape drug action, detection, and clinical risk—an angle that has been largely absent from previous reviews.

### 7.4. Comparison with Previous Reviews and Added Value of Present Study

While several earlier review articles have addressed various aspects of mephedrone, ranging from its pharmacology and clinical presentations to its abuse potential and neurotoxicity, none has provided an integrative framework centered on metabolic processes or critically examined how these processes modulate clinical and forensic outcomes. For example, the seminal review by Schifano et al. (2010) [32] focused on the emergence of mephedrone in recreational settings, its legal classification, its modes of intake, user-reported effects, and early cases of intoxication or death [32]. However, that review, while timely and informative for its era, was constrained by the limited analytical and clinical data available at the time and did not consider the metabolic dimension or stereochemical complexity of mephedrone. A 2016 analysis of mephedrone’s neurotoxicity mechanisms compared the compound with methamphetamine, noting shared pharmacodynamic properties such as dopamine transporter interactions and hyperthermic effects. Intriguingly, it reported that, despite such similarities, mephedrone does not appear to produce lasting dopaminergic neurotoxicity in animal models [33]. However, the proposed hypothesis that metabolism might explain this discrepancy remained untested, and the authors did not explore metabolic pathways or metabolite-specific toxicities. In 2020, a comparative review explored the similarities and differences between mephedrone and MDMA in terms of their psychostimulant and empathogenic effects, patterns of use, neurocognitive risks, and public health impacts. It concluded that mephedrone may have lower neurotoxicity than MDMA but greater abuse potential due to its shorter duration of action and binge-inducing profile [34]. Despite offering valuable behavioral comparisons, this review omitted a discussion of metabolic processes altogether, leaving a critical mechanistic gap in the interpretation of the observed outcomes. The most recent contribution in this area, a 2025 systematic review of blood mephedrone concentrations in fatal and non-fatal intoxication cases, provides robust forensic toxicological data. It demonstrates a statistically significant relationship between higher blood concentrations and fatal outcomes and suggests potential thresholds for clinical risk stratification [35]. However, while analytically rigorous, this review offers no discussion of the pharmacokinetic or metabolic factors contributing to interindividual variability in blood levels or outcomes, again limiting the translational utility of its findings. In contrast, the current review fills these knowledge gaps by synthesizing data across analytical toxicology, pharmacology, stereochemistry, and environmental science to construct a metabolism-centered framework. We emphasize the significance of stereoselective pathways, the role of CYP2D6 polymorphisms, and the biological activity of downstream metabolites such as normephedrone and its dicarboxylic acid conjugates (e.g., N-succinyl-, N-glutaryl-, and N-adipoyl-normephedrone). By doing so, we connect specific metabolic events with clinical presentations, toxicodynamic effects, and forensic detectability, offering mechanistic explanations that earlier reviews have not addressed. Furthermore, by incorporating environmental monitoring data and discussing metabolite stability in wastewater, we add an epidemiological perspective that has been largely absent from previous reviews. This convergence of molecular, clinical, and population-level insights enables more accurate risk assessment and supports the design of targeted interventions. Overall, this manuscript not only updates the field with the latest data on mephedrone biotransformation but also addresses important clinical and forensic implications of metabolite variability. In doing so, it advances the field from descriptive aggregation to mechanistically informed synthesis.

### 7.5. Limitations and Future Perspectives

Collectively, these findings indicate that the metabolic profile of mephedrone significantly influences its pharmacological and toxicological properties and opens avenues for improved detection strategies [11]. However, the interspecies differences observed in preclinical models underscore the need for further clinical research to validate these findings in humans [12]. A key limitation across existing studies is the lack of standardized methods for the quantification of enantiomer-specific metabolites in clinical and forensic samples. This hinders the ability to correlate metabolic profiles with clinical severity or toxic outcomes. In particular, many studies fail to distinguish between the R- and S-enantiomers of mephedrone and its metabolites, despite their differing neurochemical effects. Future investigations should focus on correlating metabolic biomarkers with clinical outcomes, taking into account interindividual variability, particularly due to genetic polymorphisms in enzymes such as CYP2D6 [13]. This is especially important given the evidence for CYP2D6-dependent metabolic pathways that could modulate both toxicity and detectability. However, the current human data are sparse, and no large-scale clinical studies have stratified participants by metabolic phenotype—an omission that limits translational relevance. In addition, integrating pharmacokinetic, toxicological, and analytical data will be crucial for a comprehensive evaluation of mephedrone’s impact on human health [14,15]. The lack of longitudinal data on metabolite persistence in various matrices (blood, urine, wastewater) also constrains efforts to establish timelines of exposure or risk windows. For instance, dicarboxylic acid conjugates may offer extended detection windows, but their long-term stability, interindividual variability, and diagnostic value remain underexplored. Finally, further studies are warranted to assess the long-term stability of these metabolites and their role in chronic toxicity, thereby informing both risk assessments and targeted therapeutic interventions [30]. Only through the development of harmonized protocols—combining stereoselective analysis, genotype-informed modeling, and real-world toxicovigilance—can the field move toward a predictive and actionable understanding of mephedrone metabolism.

## 8. Conclusions

Overall, while significant progress has been made in characterizing the metabolism of mephedrone, further clinical studies are imperative to fully elucidate these pathways, refine risk assessments, and enhance the therapeutic management of mephedrone-related toxicity. The continued integration of multidisciplinary research efforts will be key to translating these findings into effective public health strategies and clinical interventions. Future investigations should include long-term studies that monitor the chronic effects of mephedrone exposure and evaluate the persistence of its metabolites over time. A more thorough understanding of the interindividual variability in metabolic responses is also essential to tailor preventive and therapeutic approaches. Additionally, standardizing analytical methodologies will improve the reproducibility and comparability of data across different research groups. It is crucial to explore the mechanisms underlying the neurotoxic effects of specific metabolites to identify potential targets for intervention. Collaborative efforts among clinicians, toxicologists, and pharmacologists will be vital in developing comprehensive management strategies. Finally, establishing robust monitoring systems for mephedrone use will help in the early detection of emerging trends and in the formulation of timely public health responses. The metabolism-centered perspective adopted in this review offers a valuable foundation for both basic and applied toxicological research. By elucidating the links between metabolic pathways and observable toxicodynamic outcomes, this work contributes to a more predictive and mechanistically grounded understanding of mephedrone’s health impacts. Importantly, by integrating evidence from pharmacology, stereochemistry, forensic science, and environmental monitoring, this review provides a cross-disciplinary framework that has been largely absent in the previous literature.

## Figures and Tables

**Figure 1 ijms-26-07656-f001:**
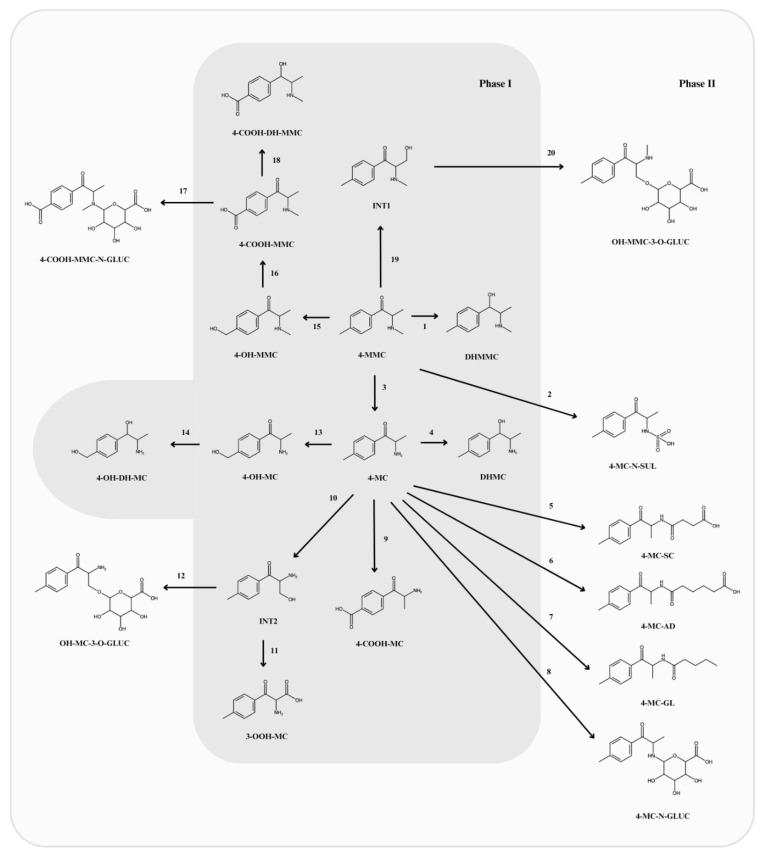
The proposed metabolic pathway of mephedrone based on the available literature. 1—reduction of the carbonyl group; 2—N-demethylation and sulfonation; 3—N-demethylation; 4—reduction of the carbonyl group; 5—conjugation with succinyl-CoA; 6—conjugation with adipoyl-CoA; 7—conjugation with glutaryl-CoA; 8—N-glucuronidation; 9—carboxylation; 10—hydroxylation; 11—oxidation; 12—O-glucuronidation; 13—hydroxylation; 14—reduction of the carbonyl group; 15—hydroxylation; 16—oxidation; 17—N-glucuronidation; 18—reduction of the carbonyl group; 19—hydroxylation; 20—O-glucuronidation.

## Data Availability

No new data were generated.

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
