# Peer review of "Mephedrone and Its Metabolites: A Narrative Review"

_ijms, 2025, doi:10.3390/ijms26157656_

Round 1
Reviewer 1 Report
Comments and Suggestions for Authors
I would like to congratulate the Authors on their work. However, I have some concerns regarding the structure of the paper.
First of all, every statement should be supported by a specific reference at the end of the sentence. For example, lines 34-35: “New psychoactive substances (NPS) have emerged as a significant global public health issue,” as well as lines 40, 42, 46, 55-56, etc. The same applies to section 3, “Preclinical Studies”; every study cited needs to be clearly referenced, for example, lines 131-139. Similarly, in paragraph 4: “Mephedrone, like other new psychoactive substances (NPS), can be detected and quantified in biological samples, including whole blood, plasma, serum, and urine.” Please meticulously check every sentence of your valuable work.
Regarding references, please avoid self-citations such as references 4 and 5.
You present this work as a narrative review. Please explain the methodology you used, including how many papers you included and excluded, and the reasons for these decisions.
I suggest you revise the overall structure of the manuscript. Avoid redundancy, try to summarize some ancillary concepts, and reorganize the discussion. Consider adding dedicated “Results” and “Conclusion” sections. Additionally, please include a discussion of the study’s limitations and future perspectives.
Author Response
Response to Reviewer #1:
Dear Reviewer nr 1,
Comment 1: “I would like to congratulate the Authors on their work. However, I have some concerns regarding the structure of the paper.”
Reply 1: Thank you for your positive comment on our manuscript. Below, we provide our responses to the suggestions and remarks you have kindly shared.
Comment 2: “First of all, every statement should be supported by a specific reference at the end of the sentence. For example, lines 34-35: “New psychoactive substances (NPS) have emerged as a significant global public health issue,” as well as lines 40, 42, 46, 55-56, etc. The same applies to section 3, “Preclinical Studies”; every study cited needs to be clearly referenced, for example, lines 131-139. Similarly, in paragraph 4: “Mephedrone, like other new psychoactive substances (NPS), can be detected and quantified in biological samples, including whole blood, plasma, serum, and urine.” Please meticulously check every sentence of your valuable work.”
Reply 2: Thank you for your constructive comment. Where sentences previously contained multiple claims or data points supported by several references, we revised the structure by dividing or rephrasing the content. This allowed us to assign individual references to specific statements in a clearer and more precise manner. We applied this approach consistently throughout the manuscript, including in the preclinical section and the parts discussing mephedrone analysis, to enhance the transparency and traceability of cited sources.
Comment 3: “Regarding references, please avoid self-citations such as references 4 and 5.”
Reply 3: Thank you for your comment regarding the use of self-citations. We fully understand the importance of maintaining objectivity and avoiding unnecessary references to our own work. However, in this case, references 4 and 5 were included not for the purpose of citation itself, but because they provide essential, original data that are highly relevant to the topic discussed. Reference 4, published in The Lancet Psychiatry, highlights the growing problem of mephedrone use in Warsaw based on long-term clinical observations. Reference 5 presents data from a unique sample of 601 hospitalized patients and offers valuable insights into psychiatric characteristics and patterns of substance use. These findings are not available in other sources and are critical for illustrating both the scale and clinical complexity of mephedrone-related issues in the Central European context. While we understand the general concern regarding self-citations, in this case we believe their inclusion is warranted due to the specificity and importance of the data they provide.
Comment 4: “You present this work as a narrative review. Please explain the methodology you used, including how many papers you included and excluded, and the reasons for these decisions.”
Reply 4: Thank you for pointing this out. We have revised the Methods section to clearly describe the search strategy, the number of included studies, and the rationale for inclusion. As this is a narrative review, we did not apply formal exclusion criteria but instead focused on the relevance of each article to the topic. We hope this clarification satisfies your request.
Comment 5: “I suggest you revise the overall structure of the manuscript. Avoid redundancy, try to summarize some ancillary concepts, and reorganize the discussion. Consider adding dedicated “Results” and “Conclusion” sections. Additionally, please include a discussion of the study’s limitations and future perspectives.”
Reply 5: In response to the Reviewer’s suggestions, we revised the structure of the manuscript to enhance clarity and coherence. The main body was reorganized into clearly labeled thematic sections to facilitate navigation and understanding. A dedicated Conclusion section was added to summarize the key findings. Additionally, we included a Limitations and Future Perspectives section to address current research gaps and propose directions for further study. Ancillary information was carefully condensed to improve focus and reduce repetition.

Reviewer 2 Report
Comments and Suggestions for Authors
The manuscript presents a review on mephedrone with a clear and well-organized structure. The writing is appropriate, and the topic is relevant given the current importance of synthetic psychoactive substance use. However, I believe the work lacks significant added value to justify its publication in its current form.
A preliminary review of the literature suggests that several existing reviews have already addressed the state of the art on this substance, some offering broader or more updated perspectives. This articles could have been cited and used by the authors to better position their work within the existing body of knowledge. These, along with many other relevant references, would allow the manuscript to more effectively reflect what has already been done in the field.
While the article is structurally sound, it does not position itself as a comprehensive review, nor does it offer a critical or novel perspective that distinguishes it from previous literature. I suggest that the authors expand their literature review, incorporate comparative analyses with existing reviews, and clearly define the manuscript’s unique contribution. In its current state, the work does not meet the expected standards of depth and originality for a review article.
Given that this is a review article, and particularly a narrative review as indicated by the title, I would have expected to see a more comprehensive contextualization of previous work on the topic. For instance, key review articles such as the ones listed below are among the most readily accessible and cited when searching terms like Mephedrone in databases or academic search engines such as Google Scholar. Including these, or discussing why they were not relevant, would strengthen the contextual framework of the manuscript:
Mephedrone (4-methylmethcathinone; ‘meow meow’): chemical, pharmacological and clinical issues | Psychopharmacology
Mephedrone concentrations in clinical intoxications and fatal cases: a systematic review | Forensic Toxicology
Mephedrone: An Overview of Its Neurotoxic Potential - ScienceDirect
Mephedrone and MDMA: A comparative review - PubMed
https://pubs.acs.org/doi/abs/10.1021/bk-2024-1481.ch006
Regarding the novelty and added value of your review, I believe the manuscript would benefit from a clearer articulation of its main contribution, especially in terms of how it differs from, updates, or synthesizes previous literature. Highlighting how this work advances the field would be essential for fulfilling the purpose of a review article.
Author Response
Response to Reviewer #2:
Comment 1: “The manuscript presents a review on mephedrone with a clear and well-organized structure. The writing is appropriate, and the topic is relevant given the current importance of synthetic psychoactive substance use. However, I believe the work lacks significant added value to justify its publication in its current form.”
Reply 1: We thank you for the positive remark and the valuable suggestion regarding the manuscript. Below, we present an explanation of the changes made based on the comments and recommendations received.
Comment 2: “A preliminary review of the literature suggests that several existing reviews have already addressed the state of the art on this substance, some offering broader or more updated perspectives. This articles could have been cited and used by the authors to better position their work within the existing body of knowledge. These, along with many other relevant references, would allow the manuscript to more effectively reflect what has already been done in the field.
While the article is structurally sound, it does not position itself as a comprehensive review, nor does it offer a critical or novel perspective that distinguishes it from previous literature. I suggest that the authors expand their literature review, incorporate comparative analyses with existing reviews, and clearly define the manuscript’s unique contribution. In its current state, the work does not meet the expected standards of depth and originality for a review article.
Given that this is a review article, and particularly a narrative review as indicated by the title, I would have expected to see a more comprehensive contextualization of previous work on the topic. For instance, key review articles such as the ones listed below are among the most readily accessible and cited when searching terms like Mephedrone in databases or academic search engines such as Google Scholar. Including these, or discussing why they were not relevant, would strengthen the contextual framework of the manuscript:
Mephedrone (4-methylmethcathinone; ‘meow meow’): chemical, pharmacological and clinical issues | Psychopharmacology
Mephedrone concentrations in clinical intoxications and fatal cases: a systematic review | Forensic Toxicology
Mephedrone: An Overview of Its Neurotoxic Potential - ScienceDirect
Mephedrone and MDMA: A comparative review - PubMed
https://pubs.acs.org/doi/abs/10.1021/bk-2024-1481.ch006
Regarding the novelty and added value of your review, I believe the manuscript would benefit from a clearer articulation of its main contribution, especially in terms of how it differs from, updates, or synthesizes previous literature. Highlighting how this work advances the field would be essential for fulfilling the purpose of a review article.”
Reply 2: We would like to sincerely thank the Reviewer for the thoughtful and constructive feedback. In response to the concern regarding insufficient contextualization of existing literature and the unclear articulation of the manuscript’s novelty, we have significantly expanded the Discussion section by adding a new paragraph titled “Comparison with previous reviews and added value of the present study.” In this updated section, we directly address the four key reviews highlighted by the Reviewer and present a chronological comparison. We begin with the 2010 article by Schifano et al., which focused on the emergence of mephedrone as a recreational drug and summarized its early clinical observations but lacked in-depth discussion of metabolism due to limited data available at the time. Next, we reference the 2016 analysis that compared mephedrone with methamphetamine and pointed out their shared pharmacodynamic effects while noting the surprising lack of persistent neurotoxicity in animal models, yet without exploring metabolic mechanisms. We also include the 2020 comparative review of mephedrone and MDMA, which examined user patterns, neurocognitive risks, and public health concerns, although it did not address pharmacokinetics or metabolite-specific contributions. Finally, we discuss the 2025 systematic review of blood mephedrone concentrations in fatal and non-fatal cases, which provided important forensic data and proposed thresholds for clinical severity but did not explore mechanistic or metabolic pathways. In contrast to these previous reviews, we explain that our manuscript offers an integrative, metabolism-focused synthesis that bridges findings from multiple disciplines such as analytical toxicology, stereochemistry, pharmacology, and forensic science. We emphasize the novelty of our approach, including detailed discussion of stereoselective metabolic transformations, implications of CYP2D6 polymorphisms, the biological activity of conjugated metabolites, and their relevance for both biomonitoring and clinical toxicology. By integrating data on environmental detection and metabolite stability, we also introduce a new epidemiological angle to the discourse on mephedrone. We believe these additions significantly improve the positioning and originality of the manuscript and directly address the Reviewer’s recommendation to clarify its value and contribution as a comprehensive narrative review.

Round 2
Reviewer 1 Report
Comments and Suggestions for Authors
Thank you for your dedication, I think that now the paper has gained clarity and scientific rigorousness reaching the level requested by the Journal.
Author Response
Dear Reviewer nr 1,
Comment 1: “Thank you for your dedication, I think that now the paper has gained clarity and scientific rigorousness reaching the level requested by the Journal.”
Reply 1: We would like to thank you for your positive feedback on our manuscript and for all the helpful suggestions you have provided.

Reviewer 2 Report
Comments and Suggestions for Authors
I appreciate the authors detailed and thoughtful response to my comments, and I acknowledge the effort made to expand the manuscript, particularly by incorporating a new section addressing previous reviews and by highlighting the intended contribution of the current work.
The added discussion comparing the manuscript with existing reviews is a valuable improvement and helps to clarify how the present review differs in its focus, particularly on metabolism, and forensic implications. This effort to better position the manuscript within the existing body of knowledge is commendable.
However, while these additions enhance the manuscript’s clarity and scope, I believe that certain concerns still remain. Specifically, although the comparison with previous literature is now more explicit, the manuscript still falls somewhat short in terms of critical synthesis. Much of the discussion is descriptive rather than analytical, and the connections between the cited disciplines (e.g., toxicology, pharmacology, stereochemistry, and forensic science) could be more cohesively articulated to support the manuscript’s central thesis.
Moreover, if the novelty of the review lies in its metabolism-centered perspective, this needs to be more strongly emphasized throughout the manuscript, not just in the discussion, by making this framing explicit in the introduction and structuring the review accordingly.
In conclusion, while the manuscript has improved and the authors have addressed many of the points raised, I believe additional refinement is needed for the article to meet the standards of a high-impact review. Strengthening the critical analysis, refining the structure to align with the proposed novel perspective, and making the added value more explicit across the manuscript would further elevate its scientific impact.
Author Response
Dear Reviewer nr 2,
Comment 1: “I appreciate the authors detailed and thoughtful response to my comments, and I acknowledge the effort made to expand the manuscript, particularly by incorporating a new section addressing previous reviews and by highlighting the intended contribution of the current work. The added discussion comparing the manuscript with existing reviews is a valuable improvement and helps to clarify how the present review differs in its focus, particularly on metabolism, and forensic implications. This effort to better position the manuscript within the existing body of knowledge is commendable. However, while these additions enhance the manuscript’s clarity and scope, I believe that certain concerns still remain. Specifically, although the comparison with previous literature is now more explicit, the manuscript still falls somewhat short in terms of critical synthesis. Much of the discussion is descriptive rather than analytical, and the connections between the cited disciplines (e.g., toxicology, pharmacology, stereochemistry, and forensic science) could be more cohesively articulated to support the manuscript’s central thesis. Moreover, if the novelty of the review lies in its metabolism-centered perspective, this needs to be more strongly emphasized throughout the manuscript, not just in the discussion, by making this framing explicit in the introduction and structuring the review accordingly. In conclusion, while the manuscript has improved and the authors have addressed many of the points raised, I believe additional refinement is needed for the article to meet the standards of a high-impact review. Strengthening the critical analysis, refining the structure to align with the proposed novel perspective, and making the added value more explicit across the manuscript would further elevate its scientific impact.”
Reply 1: We would like to sincerely thank the Reviewer for the careful reading of our manuscript and for the insightful comments provided throughout the review process. We are grateful for the recognition of the improvements made in response to the previous round of suggestions, particularly the additions regarding comparison with prior literature and clarification of the manuscript’s intended contribution. Your feedback has been instrumental in guiding substantial further revisions that we believe have significantly strengthened the manuscript’s coherence, analytical depth, and overall scientific value. In this revised version, we have thoroughly addressed the remaining concerns, especially those pertaining to the need for deeper critical synthesis, stronger articulation of interdisciplinary connections, and more consistent emphasis on the metabolism-centered perspective that underpins our review. Specifically, we have extensively revised the discussion section to move beyond description and include more analytical and evaluative commentary. For example, we now highlight inconsistencies across the literature regarding CYP2D6-related stereoselectivity and the translational consequences of divergent metabolic findings. We critically examine methodological limitations, such as the continued use of GC-MS despite known degradation risks, and emphasize how this impairs diagnostic reliability. Where neurotoxicity data conflict, we note the lack of standardized behavioral endpoints and the need for longitudinal designs. These additions reflect a deliberate effort to synthesize data across studies, assess their limitations, and extract mechanistic insight, rather than simply catalog results. To reinforce the interdisciplinary integration, we have added language that explicitly connects stereoselective pharmacology with enzyme polymorphism, forensic detection challenges, and public health surveillance. The discussion now closes with a paragraph that explicitly positions these interactions within a unified metabolism-centered framework, showing how the convergence of toxicokinetics, stereochemistry, analytical toxicology, and epidemiology enables more informed interpretation of mephedrone’s health risks. We also responded to your suggestion to emphasize the novelty of the metabolism-centered approach more consistently throughout the manuscript. In the introduction, we now clearly state that the structure and intent of the review are built around metabolic pathways as the organizing principle, something not previously done in this area. This conceptual framing is reiterated in the revised discussion and conclusions, where we clarify that the review offers not only a descriptive account of metabolism, but a translationally oriented framework linking metabolic processes to drug effects, detection strategies, and clinical outcomes. Finally, in line with your advice, we have made the added value of the present review more explicit in multiple sections. In the discussion, we now directly compare our work to earlier reviews, pointing out the absence of metabolic synthesis in prior literature and showing how our work fills this gap by integrating data across molecular, clinical, and environmental levels. In the conclusion, we highlight the cross-disciplinary nature of our approach and its implications for toxicological research, risk stratification, and monitoring strategies. We appreciate the opportunity to revise the manuscript in light of your detailed guidance. Your feedback has substantially improved the clarity, critical depth, and scientific coherence of the review. We hope the changes now fully address your concerns and meet the expectations for publication in a high-impact journal.

Round 3
Reviewer 2 Report
Comments and Suggestions for Authors
No additional comments.